# ChatGPT and Bard in Plastic Surgery: Hype or Hope?

Ania Labouchère * 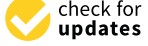 and Wassim Raffoul 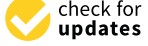

Plastic, Reconstructive and Hand Surgery Service, Lausanne University Hospital, University of Lausanne, Rue du Bugnon 46, CH-1011 Lausanne, Switzerland; wassim.raffoul@chuv.ch
* Correspondence: ania.labouchere@chuv.ch

**Abstract:** Online artificial intelligence (AI) tools have recently gained in popularity. So-called "generative AI" chatbots unlock new opportunities to access vast realms of knowledge when being prompted by users. Here, we test the capabilities of two such AIs in order to determine the benefits for plastic surgery while also assessing the potential risks. Future developments are outlined. We used the online portals of OpenAI's ChatGPT (version 3.5) and Google's Bard to ask a set of questions and give specific commands. The results provided by the two tools were compared and analyzed by a committee. For professional plastic surgeons, we found that ChatGPT and Bard can be of help when it comes to conducting scientific reviews and helping with scientific writing but are of limited use due to the superficiality of their answers in specific domains. For medical students, in addition to the above, they provide useful educational material with respect to surgical methods and exam preparation. For patients, they can help when it comes to preparing for an intervention, weighing the risks and benefits, while providing guidance on optimal post-operative care. ChatGPT and Bard open widely accessible data to every internet user. While they might create a sense of "magic" due to their chatbot interfaces, they nonetheless can help to increase productivity. For professional surgeons, they produce superficial answers—for now—albeit providing help with scientific writing and literature reviews. For medical students, they are great tools to deepen their knowledge about specific topics such as surgical methods and exam preparation. For patients, they can help in translating complicated medical jargon into understandable lingo and provide support for pre-operative as well as post-operative care. Such AI tools should be used cautiously, as their answers are not always precise or accurate, and should always be used in combination with expert medical guidance.

**Keywords:** artificial intelligence; plastic surgery; Bard; ChatGPT

## 1. Introduction

In late 2022, the US-based research laboratory OpenAI released ChatGPT (the acronym GPT standing for Generative Pre-Trained Transformer) [1]. Technically, it is a conversation large language model (LLM) with an underlying technology known as generative artificial intelligence (GAI). The GAI architecture uses deep learning methods to generate natural language text after the model has been trained on myriads of text data such as books, websites and articles. Answers are provided when prompted by a user on an online platform.

Similarly, in February 2023, Google rolled out Bard, the direct competitor of Chat-GPT [2]. It is a next-generation language with conversation capabilities powered by the firm's Language Model for Dialogue Applications (LaMDA).

With rapidly evolving capacities, thought leaders posit that such tools "will redefine human knowledge, accelerate changes in the fabric of our reality, and reorganize politics and society" [3]. With vast amounts of data becoming available, queryable and organized into understandable information to humans prompting these chatbots, success was instant, with over 100 million users two months after the launch of ChatGPT [4].

ChatGPT was trained with data from up to 2021, while Bard continually draws information from the internet [5]. These "revolutionary demonstration[s] of technology" [6] have

unlocked numerous applications ranging from writing essays and articles [7] to correcting computer code [8], drafting job applications [9] and passing exams [10], among others.

In the medical field, more specifically surgery, ChatGPT and Bard have also spurred interest, while authors advocate for precaution upon using the tool [11–18]. Many advantages are noted, such as a reduction in the time spent on literature searches and reviews, data extraction and analysis, academic writing of research reports, etc. Simply put, these would lead to better and more informed decision-making processes and ultimately, better treatments and outcomes for patients. It has also been suggested that such chatbots could help in enhancing surgical education techniques and related education [19–23]. Chatbot assistants can also automate repetitive tasks such as discharge summaries and patient letters, thereby avoiding possible human error and saving precious time to allow clinicians to spend extra moments with patients [24,25]. From a patient perspective, ChatGPT can also respond interactively and accurately to questions arising from a patient with respect to a condition or procedure [26].

On the flip side, the use of this technology should be carried out with critical limitations in mind. In effect, ChatGPT has been trained on data up to 2021, which is self-limiting in itself to incorporate the most up-to-date scientific outcomes. Also, the chatbot, while generating grammatically correct answers and "scientifically plausible text", can deliver relatively superficial answers lacking in precision and depth [27]. Going further, other studies have unveiled that GAIs have completely made up answers and references ("neural hallucinations"), which underlines the fact that they are prone to errors (e.g., references, interpretation) and can miss, overlook or misinterpret information (e.g., relevant studies, medical nuances), let alone be biased [28,29]. This caveat is crystallized in a recent article by Ollivier, who argues that an AI chatbot can be listed as a co-author based on the absence of responsibility for the article's claims [30].

How can ChatGPT and Bard be of use in the field of plastic surgery? To answer this question, we have identified three groups in this field: (i) professional plastic surgeons with several years of experience, (ii) medical students training to become plastic surgeons and (iii) patients undergoing plastic surgery. Here, we report on the use of ChatGPT and Bard by these three groups of users and identify opportunities while providing an analysis for specific use cases.

In this context, the objective is to assess the relevance of two AIs (ChatGPT and Bard) to complete common tasks of three groups of persons in the medical field outlined above, as few comparative evaluations have been performed to date.

## 2. Materials and Methods

### 2.1. Choice of AI

We selected ChatGPT and Bard due to the prominence of these models in 2023, representing two significant advancements by OpenAI and Google, respectively, in natural language understanding and dialogue capabilities.

### 2.2. Writing of Questions

In total, 27 questions were created based on facial aging and facelifts and divided into three categories with nine questions each: specialists, students and patients.

For specialists, we investigated whether the GAIs can help to conduct scientific research (e.g., literature writing, review, etc.), write medical letters and support with marketing tasks (Table S1). For students, we looked at whether the two GAIs are able to assist in teaching, both for written and oral exams (Table S2). For patients, we sought to determine whether ChatGPT and Bard are useful in terms of popularizing medical concepts, providing pre- and post-operative support, particularly in detecting potential complications, and whether they are capable of knowing their own limitations by referring the patient to a specialist (Table S3).

Practically, we started by indicating our role at the beginning of every conversation ("I am a specialist", "I am a medical student", "I am a patient").

### 2.3. Testing of ChatGPT and Bard

Freely accessible (default) versions of ChatGPT 3.5 and Bard were used to conduct the study in May 2023.

### 2.4. Analysis of Responses

All questions and answers were copied and pasted into tables (see Supplementary Tables S1–S3). Responses were compared and discussed by a committee including two patients, two medical students and two specialists in plastic and reconstructive surgery. A general score was attributed to every answer based on quality (0 = no answer, 1 = poor, 2 = acceptable, 3 = good, 4 = very good; a star * after the score indicates that wrong or invented elements were suggested in the answers).

## 3. Results

The answers of the two aforementioned AIs are given a few seconds after having been prompted by a user (Tables 1–3).

**Table 1.** Analysis of the answers given by ChatGPT and Bard concerning specialists by the committee.

|  | Tasks—Specialists | ChatGPT | Bard |
|---|---|---|---|
| 1.1 | Writing of letters | 4 | 3 * |
| 1.2 | Generation of medical content | 4 | 2 * |
| 1.3 | Analysis of risk factors for a procedure | 2 | 3 |
| 1.4 | Performing a literature review | 0 | 0 |
| 1.5 | Writing of a scientific article (literature review) | 0 | 0 |
| 1.6 | Performing a critical analysis of an article | 0 | 1 * |
| 1.7 | Summary of a scientific article | 4 | 3 |
| 1.8 | Citing authors | 0 | 0 * |
| 1.9 | Providing marketing material | 2 | 4 |

0 = no answer, 1 = poor, 2 = acceptable, 3 = good, 4 = very good; a star * after the score indicates that wrong or invented elements were suggested in the answers.

**Table 2.** Analysis of the answers given by ChatGPT and Bard concerning students by the committee.

|  | Tasks—Students | ChatGPT | Bard |
|---|---|---|---|
| 2.1 | Physiopathology | 3 | 3 |
| 2.2 | Anatomy | 3 | 4 |
| 2.3 | Explanation of surgical techniques | 4 | 4 |
| 2.4 | Exams: Creation of multiple-choice questions (MCQ) | 4 | 4 |
| 2.5 | Exams: Creation of a clinical case for EBOPRAS (European Board of Plastic, Reconstructive, and Aesthetic Surgery) | 4 | 2 |
| 2.6 | Exams: Questions for oral exams (EBOPRAS) | 4 | 3 |
| 2.7 | Exams: Answers of multiple-choice questions | 2 | 4 |
| 2.8 | Exams: Justifying an answer | 3 | 4 |
| 2.9 | Evaluation of surgical level | 0 | 0 |

0 = no answer, 1 = poor, 2 = acceptable, 3 = good, 4 = very good.

**Table 3.** Analysis of the answers given by ChatGPT and Bard concerning patients by the committee.

|  | Tasks—Patients | ChatGPT | Bard |
|---|---|---|---|
| 3.1 | Physiopathology | 4 | 4 |
| 3.2 | Questions about surgical techniques | 4 | 4 |
| 3.3 | Outline complications of a surgery | 4 | 4 |
| 3.4 | Present the contraindications of a surgical intervention | 4 | 4 |
| 3.5 | Visual rendering of a surgery | 0 | 0 |
| 3.6 | Suggest references of surgical drawings | 0 | 0 |
| 3.7 | Provide medical advice | 4 | 4 |
| 3.8 | Provide medical advice and opinion | 4 | 3 |
| 3.9 | Provide post-operative care advice | 4 | 4 |

0 = no answer, 1 = poor, 2 = acceptable, 3 = good, 4 = very good.

### 3.1. Specialists

#### 3.1.1. Writing of Letters

ChatGPT and Bard were capable of generating letters, including letters to insurance companies. Both generated a layout with fields to fill in (e.g., patient's name and address, if there are any allergies, etc.). Both GAIs could design, based on the given information, coherent and relevant symptoms related to the pathology. Compared to Bard, ChatGPT wrote a more complete letter that can easily be used as is, while Bard extrapolated and introduced elements that did not correspond to the indications given about the patient (e.g., "she has even stopped going to work because he is so ashamed of his appearance").

#### 3.1.2. Generation of Medical Content

We asked ChatGPT and Bard to write an anamnesis on a patient known to suffer from Ehlers–Danlos disease and who had undergone a facelift. Impressively, ChatGPT was able to generate an anamnesis based on succinct medical information, accurately highlighting the risks associated with the patient's comorbidities, without being guided by the information we provided. However, Bard, once again, invented non-relevant content (e.g., patient's name, history of aesthetic medicine, childhood medical history, social history, etc. such as, for example: "The patient was born at term and had an uneventful delivery. She was a healthy child and did not have any major illnesses or injuries. She attended college and graduated with a degree in business. She is married and has two children. She is a stay-at-home mom.").

#### 3.1.3. Analysis of Risk Factors for a Procedure

Both GAIs were able to identify relevant risk factors in a medical history and provide explanations. Without being directly specified by the prompt, the GAIs spontaneously considered the end user as a patient rather than a specialist and recommended that the patient consult a specialist to assess the risks and benefits of such an intervention.

#### 3.1.4. Performing a Literature Review

Both GAIs did not produce a literature review in their free versions. ChatGPT specified the following: "I don't have direct access to external sources such as medical databases or journals to conduct a literature search", while Bard indicated that: "I'm unable to help you with that, as I'm only a language model and don't have the necessary information or abilities." Nevertheless, ChatGPT spontaneously explained which surgical techniques have gained in popularity while stating that it is based on knowledge prior to 2021.

### 3.1.5. Writing of a Scientific Article (Literature Review)

Both GAIs created a succinct summary with introduction, history, techniques and complications rather than a literature review. ChatGPT concludes with research perspectives, while Bard elaborates what it calls a "literature review" by vaguely describing two studies without citing either title or authors. Moreover, no sources were cited for either.

### 3.1.6. Perform a Critical Analysis of an Article

On the one hand, by indicating the Digital Object Identifier (DOI) of a research paper, ChatGPT was unable to access it and did not critically analyze the article. On the other hand, Bard was able to access the article via the internet. However, the latter mentioned general study limitations (small sample size, language bias, short follow up, etc.) without being able to highlight the limitations cited in the discussion section of the article. Bard also mentioned incomplete information (it lists 11 articles used in an article, whereas there are 152).

### 3.1.7. Summary of a Scientific Article

Both GAIs could summarize an abstract of a scientific article, highlighting the study's background, results and conclusion. ChatGPT was more precise, citing the type of study (systematic review in this case) as well as a summary of the methodology used by the authors. Bard, on the other hand, added a link to the source of the article in question and its DOI, unlike ChatGPT.

### 3.1.8. Citing Authors

ChatGPT was unable to cite authors, while Bard only cited non-existent, fictitious authors. Note that when we told Bard that the list of authors was wrong, it could not find the correct list of authors. But once we had told Bard who the authors of the article were and asked it again in the same conversation (a dedicated "chat"), it was then able to quote them correctly. That being said, it would not be able to quote the authors correctly in a new conversation.

### 3.1.9. Provide Marketing Material

ChatGPT offered a structure for marketing posts by making a list of points that a specialist can develop (definition of the procedure, benefits, considerations, what to expect during the procedure and recovery) for a professional or personal blog or social media platform, while Bard directly generated usable content (definition of the procedure, risks, benefits, price, expectation).

### *3.2. Students*

### 3.2.1. Physiopathology

The two GAIs were similar in their responses. The answers explained pathophysiology in a global way without going into anatomical details. Bard added, without being asked, the risk factors leading to pathologies and how to avoid them.

### 3.2.2. Anatomy

Anatomical explanations were accurate but succinct. Bard provided a little more anatomical details, although this remained summarized and did not spontaneously cite references.

### 3.2.3. Explanation of Surgical Techniques

The techniques were listed in a summarized format. Upon being prompted with a question about which facelifts exist, ChatGPT listed the different types of facelifts, while Bard cited the different modalities (facelift, Botox, laser, etc.) that could lead to a lifting effect.

### 3.2.4. Exams: Creation of Multiple-Choice Questions (MCQ)

When asked to create MCQs, ChatGPT spontaneously created three questions with four possible choices but only one true proposition and Bard came up with (i) two questions with four possible choices (one being "All of the above") and (ii) one with three possible choices. However, the questions were basic for a specialist but seemed suitable for a medical student.

### 3.2.5. Exams: Creation of a Clinical Case for EBOPRAS (European Board of Plastic, Reconstructive, and Aesthetic Surgery)

On the one hand, ChatGPT described a clinical case that could be useful in order to review for an oral exam. It described the steps to be followed, such as history-taking, physical examination, proposed treatment and intervention, while recalling the objectives and key points to remember (from a physician's point of view) for this type of examination. Bard, on the other hand, described a case more succinctly and concluded by explaining why an EBOPRAS-certified surgeon can be of benefit to a patient.

### 3.2.6. Exams: Questions for Oral Exams (EBOPRAS)

Both AIs were capable of generating questions that can be asked during an oral exam. ChatGPT created a single but more complete question with a paragraph on the expected analysis of the case. Bard asked five brief questions without providing extra material.

### 3.2.7. Exams: Answers of Multiple-Choice Questions

Both GAIs were asked three questions. Bard answered correctly all three, while ChatGPT made an anatomical error.

### 3.2.8. Exams: Justifying an Answer

ChatGPT gave a correct but brief explanation. Bard expanded its answer a little more while addressing a patient and redirecting him/her to a specialist.

### 3.2.9. Evaluation of Surgical Level

The answers were not satisfactory and only provided from the perspective of patients.

### *3.3. For Patients*
### 3.3.1. Physiopathology

The questions were well adapted to the patient's level, with an explanation of risk factors and their physiological consequences (e.g., that the sun's UV rays induce aging through "the breakdown of collagen and elastin, which are proteins essential for maintaining skin elasticity. This can result in wrinkles, fine lines, age spots and uneven skin tone": ChatGPT) and primary prevention.

### 3.3.2. Questions about Surgical Techniques

ChatGPT provided a simple description of what a procedure is (in this case, a deep plane lift). It went on to describe the surgical incisions and what the expected results are with this type of procedure. It concluded by saying that not all patients can be candidates, and that an experienced plastic surgeon could provide more information. Bard provided an overview similar to ChatGPT, including the following elements: type of anesthesia, operation time, recovery, complication.

### 3.3.3. Outline Complications of a Surgery

The explanations and percentages given were correct. Bard went one step further by directing the patient to find a board-certified surgeon.

### 3.3.4. Present the Contraindications of a Surgical Intervention

The contraindications were satisfactorily listed.

### 3.3.5. Visual Rendering of a Surgery

Both GAIs were unable to generate images. ChatGPT nevertheless verbally explained the incisions to be performed.

### 3.3.6. Suggest References of Surgical Drawings

Both GAIs were unable to communicate specific website references or direct links to sources containing drawings of surgical operations.

### 3.3.7. Provide Medical Advice

By pointing out symptoms, both GAIs were able to indicate the possible pathology as well as its differential diagnoses to the patient. They also suggested that the patient consult a specialized surgeon.

### 3.3.8. Provide Medical Advice and Opinion

ChatGPT clearly indicated that it is an AI, which is not capable of making a diagnosis. It evoked hypotheses and told the patient to consult his/her doctor. Bard, on the other hand, said it is difficult to make a diagnosis with little information. It also evoked hypotheses and told the patient to consult a specialist.

### 3.3.9. Provide Post-Operative Care Advice

Both GAIs provided indications for post-operative care. ChatGPT initially referred to the doctor's indications, whereas Bard did not.

## 4. Discussion

Below, we consider every individual group, namely, specialists, students and patients, and infer insights about the relevance of GAIs in their roles for the questions that were prompted to both ChatGPT and Bard.

### *4.1. Specialists*

GAIs are a relevant tool for specialists in several respects, which are covered in the below sub-sections.

### 4.1.1. Medical Letters

According to DiGiorgio et al., AIs are ready to tackle many administrative tasks [14]. We have observed that Bard and ChatGPT are powerful tools for creating medical letters, as was already pointed out by Ali et al. [25]. In effect, these can, for example, be addressed to insurance companies for reimbursement purposes. We have witnessed that both can generate a template with fields to be filled (e.g., patient's name and address, if there are any allergies, etc.), which offers the advantage of synthesizing information in a structured manner. This can then easily be checked by the patients themselves, facilitating error correction and thus, saving time and money. Furthermore, based on the information given, both AIs suggest coherent, relevant symptoms linked to the pathology. Upon comparing the two, ChatGPT writes a more complete letter that can easily be used as is, while Bard extrapolates and introduces inaccurate informative elements that do not match the indications given about the patient nor his/her history.

### 4.1.2. Literature Review and Scientific Writing

While AIs can provide ideas for articles to be written, as described by Gupta et al., they are not capable of writing a scientific article, as they lack integrity and accuracy in their content, as pointed out by Alkaissi et al. [12,31]. Furthermore, as described by Ollivier et al., they are in fact unable to design experiments or formulate research hypotheses, contrary to what some researchers state in their articles [13,28,32]. In addition, when asked to cite authors, ChatGPT said it cannot, while Bard listed false authors. Surprisingly, high levels of "hallucination" or "confabulation" have been observed in the content that was

generated and have been defined as "mistakes in the generated text that are semantically or syntactically plausible but are in fact incorrect or nonsensical" [33].

As described in the results, by indicating the Digital Object Identifier (DOI) of a research paper, ChatGPT is, on the one hand, not able to access it and does not critically analyze the article. On the other hand, Bard is able to access the article. However, the latter mentions general study limitations (small sample size, language bias, short follow up, etc.) without being able to highlight the limitations cited in the discussion section of the article. Bard also mentions incorrect information. As already described by Hassan et al., generating a summary of an article is possible for both Ais, with more precision in ChatGPT [32].

To date, in its free version, ChatGPT has stopped producing literature reviews and Bard does not review the literature either. It is to be hoped that future versions will once again include a reliable and accurate review of the literature, as this would be very useful for specialists.

### 4.1.3. Assistive Support for Operations

Contrary to the predictions of Hassan et al., AIs are not yet capable of guiding specialists in their care before (choice of surgery), during (in the operating room) and after (post-operative care) and are still a long way from being able to help them in real time during surgical interventions in the operating room [32]. Indeed, as ChatGPT and Bard are not capable of reviewing the literature, the specialist will not be able to rely on the existing literature via AIs to make a treatment decision. As these AIs are not yet sufficiently powerful, they are not capable of reasoning. During surgery, AIs are not capable of guiding the surgeon by voice or video, as they cannot yet analyze voice or video messages in real time. Post-operatively, however, AI can provide support for the patient (see "Patients" section), saving the specialist time. However, it will not be able to guide the specialist in his or her choices, clinical acumen and management.

### 4.1.4. Media Creation

The application where AIs can also be effective for specialists is in the creation of media content, as indicated by Gupta et al. Bassiri-Tehran specifies that ChatGPT is capable of creating content ideas, captions, blogs, general content and newsletters for email or social media, which Bard can also perform [12,34].

### *4.2. Students*

Medical Teacher and Exam Preparation

A small number of studies have already attempted to evaluate the ability of ChatGPT to teach medicine. Kung et al. demonstrated that ChatGPT approaches or exceeds the pass mark for the United States Medical Licensing Examination (USMLE) as of February 2023 [22]. Again in the US, ChatGPT achieves the level of a first-year surgeon based on the Plastic Surgery Inservice Training Examination (PSITE) resident assessment tool. Namkee Oh et al. showed that ChatGPT achieves a 76.4% pass rate on the Korean General Surgery Board Examination [19,35]. Two authors have shown that ChatGPT outperforms Bard in radiology MCQs, with 87.11% correct answers for ChatGPT versus 70.44% for Bard in Patil's American College of Radiology's Diagnostic Radiology In-Training with 380 questions, and 65.0% for ChatGPT versus 38.8% for Bard in Toyama's Japan Radiological Society's MCQ with 103 questions [36].

With regard to learning from multiple-choice questions (MCQ) and reviewing clinical cases, both ChatGPT and Bard are capable of generating MCQs aimed at medical school students or beginner surgeons with appropriate levels of complexity. Clinical cases are better developed in ChatGPT, with a structure closer to an oral exam, as opposed to Bard, which simply lists elements. It should be noted that on the three MCQ questions prompted to the AIs, ChatGPT made an anatomical error, while Bard answered correctly. Agarwal et al. asked chatbots to produce 110 MCQs based on the physiological knowledge required for a bachelor's degree in medicine and a bachelor's degree in surgery in India [37]. They

then assessed the validity of the questions produced and the level and ability of reasoning required to answer them. ChatGPT and Bard scored similarly on all three criteria [37,38].

Based on our questions, we can deduce that ChatGPT and Bard can be learning aids for medical students and novice surgeons. However, they cannot support the education of advanced surgeons due to their limitations in terms of content, inability to think critically and hypothesize, and analysis. When it comes to learning about pathophysiology and anatomy, the AIs remain superficial and do not provide any sources, making it impossible to learn in depth and with precision. In this regard, Sevgi et al. point out that it is not advisable to rely solely on ChatGPT as an educational resource [23].

The same applies to surgical techniques. AIs provide a general overview but do not allow detailed training by a more experienced surgeon. In effect, surgery is primarily a manual job where precise hand movements must be taught by experienced teachers with extensive experience in the field.

### *4.3. Patients*

4.3.1. General Support

For patients, the AIs we have looked at seem perfectly adapted and equivalent in terms of explaining procedures, indications and contraindications, complications and post-operative follow-up. Overall, we noted that Bard developed its answers a little more.

A study by Ayers et al. showed that, out of 195 exchanges, 78.6% of patients preferred the responses of the AIs to those of the surgeons because they were considered of higher quality and more empathetic [39]. As the authors suggest, one solution would be to create a chatbot on an online portal, edited directly by doctors, that would answer the questions of patients. Gupta et al. also posit that GAIs could be used to support the post-operative period between specialist appointments [12].

The downside is that, currently, neither ChatGPT nor Bard AIs are able to create visual drawings to explain the interventions. The fact that AIs cannot generate drawings limits personalized communication with patients. Unfortunately, they are also unable to provide sources (e.g., books, videos, presentations slides, etc.) to make up for this shortcoming. In a profession such as surgery, there are many different surgical techniques and every one is operator-dependent. Visual content provides patients with a visual medium to understand medical content, students with a varied learning medium and specialists with the opportunity to create content for research or marketing.

4.3.2. Medical Advice

Upon asking for medical advice, ChatGPT seems more reliable and always refers patients to a specialist, whereas Bard does not do so systematically. We could conclude that ChatGPT's support seems secure, as it acknowledges its limitations by referring patients to surgeons, indicating to consult them pre-, per- and post-operatively [40].

### *4.4. General Considerations*

4.4.1. Limitation

Our main limitation is the exclusion of other GAIs (e.g., Perplexity AI, Bing AI, etc from the study. A more comprehensive analysis might have involved comparing a broader range of AI models to provide a more nuanced understanding of the capabilities and limitations of different conversational AI systems in medical contexts.

4.4.2. Performance

From a chronological perspective, Bard provided answers more quickly than ChatGPT, as its answers were in general shorter, as also observed by Mayank and Patil. This represents a time-saving feature that is beneficial for the end user [36,37].

### 4.4.3. Data Privacy

Patient-related data should follow strict data privacy guidelines to avoid leaks or subsequent use for unrelated matters.

### 4.4.4. Political Environment and Regulations

Politically, GAIs have accelerated the discussions about their regulation, most notably in the European Union (EU). In effect, as proposed in the EU AI Act, GAIs would have to comply with transparency requirements such as (i) disclosing that the content was generated by AI, (ii) designing the model to prevent it from generating illegal content and (iii) publishing summaries of copyrighted data used for training [41].

To date, no AI tool is able to guarantee the confidentiality of the data fed into its system. Since AIs are formatted to enrich themselves (and thus memorize and restitute) with the content they are trained on, they could divulge it. For the time being, therefore, it is illegal in Switzerland and Europe to feed AIs personal data. Users must therefore take care to anonymize the data transmitted to the AI [42].

Globally, in November 2023, a first international agreement was signed (Bletchley declaration) on GAI safety. Organized in London, this summit will be held once a year, bringing together experts and politicians to discuss the risks and regulation of AI.

### 5. Conclusions

In a clinical setting, ChatGPT and Bard have proven to be efficient assistants with respect to specific tasks. However, in order to avoid several shortcomings and pitfalls that were encountered, it is recommended to use both ChatGPT and Bard (as well as other similar GAI chatbots) responsibly and with strict guidelines (e.g., verification of sources, critical analysis of answers, awareness of risks pertaining to data privacy, etc.). It is worth noting that the answers provided by ChatGPT and Bard are always given in response to an original query provided by the user. Therefore, the creativity and ability of a user can unlock new, better answers and overcome existing limitations. As pointed out by Kissinger, Schmidt and Huttenlocher: "Inherently, highly complex AI furthers human knowledge but not human understanding—a phenomenon contrary to almost all of post-Enlightenment modernity. Yet at the same time AI, when coupled with human reason, stands to be a more powerful means of discovery than human reason alone." Furthermore, the constant evolution of training data and underlying technologies implies that the quality of answers provided should increase with time.

**Supplementary Materials:** The following supporting information can be downloaded at: https://www.mdpi.com/article/10.3390/surgeries5010006/s1, Table S1. Specialists, Table S2. Students. Table S3. Patients.

**Author Contributions:** Conceptualization, methodology, software, A.L.; validation, A.L. and W.R.; formal analysis, investigation, resources, data curation, A.L.; writing—original draft preparation, writing—review and editing, A.L.; supervision, W.R. All authors have read and agreed to the published version of the manuscript.

**Funding:** This research received no external funding.

**Institutional Review Board Statement:** Not applicable.

**Data Availability Statement:** Data is contained within the article and supplementary material.

**Acknowledgments:** The authors would like to thank Philippe Labouchère for his help with using ChatGPT and Bard, as well as for editing and revising the manuscript.

**Conflicts of Interest:** The authors declare no conflicts of interest.

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
