# Peer review of "ChatGPT and Bard in Plastic Surgery: Hype or Hope?"

_2673-4095, doi:10.3390/surgeries5010006_

Round 1
Reviewer 1 Report
Comments and Suggestions for AuthorsØ The paper discusses the use of AI in medical contexts, specifically comparing ChatGPT and Bard in plastic surgery. How do the authors justify the selection of these two AI models for their study, and are there any other AI models that could have been considered for a more comprehensive analysis?
Ø The paper mentions the inability of ChatGPT and Bard to create visual drawings or provide sources for surgical operations. How significant is this limitation in the context of the study's objectives, and are there potential ways to overcome this limitation?
Ø In the analysis of risk factors for a procedure, the paper notes that both AI models were able to identify relevant risk factors. How does this capability compare to traditional methods used by medical professionals, and what are the implications for the future use of AI in medical diagnostics?
Ø The paper evaluates the performance of ChatGPT and Bard in generating medical content and advice. How do the authors assess the accuracy and reliability of the information provided by these AI models, and what are the potential risks associated with their use in a clinical setting?
Ø The study includes a comparison of the AI models' responses to various medical scenarios. How were these scenarios chosen, and do they adequately represent the range of challenges faced in plastic surgery?
Ø The paper discusses the generation of medical content by AI. How do the authors address the ethical considerations and potential misinformation risks associated with AI-generated medical content?
Ø The paper mentions the use of a committee, including medical students and specialists, to evaluate the AI responses. How was the evaluation process structured, and what criteria were used to assess the quality of the AI responses?
Ø The paper notes the limitations of AI in performing literature reviews and scientific writing. How do the authors envision the role of AI in academic research, given these limitations?
Ø The study explores the use of AI for providing postoperative care advice. How does the advice generated by AI compare to standard postoperative care guidelines, and what is the potential for AI to supplement or enhance patient care?
Ø The paper discusses the potential for AI to save time and money in medical contexts. Can the authors provide more details on the specific areas where AI can offer the most significant cost and time efficiencies?
Author Response
Dear Reviewer, thank you for your thoughtful comments.
Ø The paper discusses the use of AI in medical contexts, specifically comparing ChatGPT and Bard in plastic surgery. How do the authors justify the selection of these two AI models for their study, and are there any other AI models that could have been considered for a more comprehensive analysis?
The selection of ChatGPT and Bard is based on their roles as AI language models with conversation capabilities (L40-42). ChatGPT, developed by OpenAI, and Bard, developed by Google, are positioned as direct competitors in the realm of conversational AI. The selection may be influenced by the prominence of these models in the AI landscape during the study period (2023) and their perceived relevance in medical applications. In response to your question, we have added the following chapter to "Materials and methods":
“2.1. Choice of AI
We selected ChatGPT and Bard due to the prominence of these models in 2023, representing two significant advancements by OpenAI and Google, respectively, in natural language understanding and dialogue capabilities. (L89-92)
and in the chapter “conclusion”:
4.4.1. Limitation
Our main limitation is the exclusion of other GAIs (e.g., Perplexity AI, Bing AI,...) from the study. A more comprehensive analysis might have involved comparing a broader range of AI models to provide a more nuanced understanding of the capabilities and limitations of different conversational AI systems in medical contexts”. (L393-397)
Ø The paper mentions the inability of ChatGPT and Bard to create visual drawings or provide sources for surgical operations. How significant is this limitation in the context of the study's objectives, and are there potential ways to overcome this limitation?
The inability of ChatGPT and Bard to create visual drawings or provide sources for surgical operations is acknowledged as a limitation of the technology in a broader context but not as a limitation of this study per se. This limitation is significant as visual aids play a crucial role in medical communication, especially in surgery. To address this limitation, future developments in AI could focus on enhancing the multimodal capabilities of models, allowing them to generate and interpret visual information in conjunction with text. The following text was added:
“Visual content provides patients with a visual medium for understanding medical content, students with a varied learning medium, and specialists with the opportunity to create content for practice, research or marketing.” (L383-385)
Ø In the analysis of risk factors for a procedure, the paper notes that both AI models were able to identify relevant risk factors. How does this capability compare to traditional methods used by medical professionals, and what are the implications for the future use of AI in medical diagnostics?
Traditional risk factor research methods vary between acquired knowledge (studies), experience and literature research. Searching for information in scientific literature can be costly, and doesn't allow everyone to access it. AI models like ChatGPT and Bard may contribute to the unification of knowledge and easier access to free information.
Ø The paper evaluates the performance of ChatGPT and Bard in generating medical content and advice. How do the authors assess the accuracy and reliability of the information provided by these AI models, and what are the potential risks associated with their use in a clinical setting?
To assess the reliability of the AIs' answers, we chose questions on subjects that are part of our medical knowledge (based on literature and experience) as plastic surgeons. We thus have the sufficient expertise as specialists to judge the relevance of the answers.
Currently, the risk of using AI in the clinic is low. This is due to the fact that the responses, which are provided by AI, are unusable in the first place because they are incomplete or lack sufficient quality.
As pointed out in the text:
“Contrary to the predictions of Hassan et al., AIs are not yet capable of guiding specialists in their care before (choice of surgery), during (in the operating room) and after (post operative care) and are still a long way from being able to help them in real time during surgical interventions in the operating room(34). Indeed, as ChatGPT and Bard are not capable of reviewing the literature, the specialist will not be able to rely on the existing literature via AIs to make a treatment decision. As these AIs are not yet sufficiently powerful, they are not capable of reasoning. During surgery, AIs are not capable of guiding the surgeon by voice or video, as they cannot yet analyze voice or video messages in real time. Post-operatively, however, AI can provide support for the patient (see "Patients" section), saving the specialist time. However, it will not be able to guide the specialist in his or her choices, clinical acumen and management.” (L317-327)
Ø The study includes a comparison of the AI models' responses to various medical scenarios. How were these scenarios chosen, and do they adequately represent the range of challenges faced in plastic surgery?
The study's scenarios aim to cover a spectrum of tasks to comprehensively assess the AI models' capabilities and limitations in different roles within the field of plastic surgery. We evaluate the AI models' performance in various aspects relevant to plastic surgery, encountered in our practice, such as research, teaching, and communication.
More information can be found in the methodology:
“27 questions were created, based on facial aging and facelifts, and divided into three categories with nine questions each: Specialists, Students and Patients.
For Specialists, we investigated whether the GAIs can help to conduct scientific research (e.g. literature writing, review, etc.), write medical letters and support with marketing tasks (Appendix 1). For Students, we looked at whether the two GAIs are able to assist in teaching, both for written and oral exams (Appendix 2). For Patients, we sought to determine whether ChatGPT and Bard are useful in terms of popularizing medical concepts, providing pre- and post-operative support, particularly in detecting potential complications, and whether they are capable of knowing their own limitations by referring the patient to a specialist (Appendix 3).
Practically, we started by indicating our role at the beginning of every conversation (“I am a Specialist”, “I am a medical student.”, “I am a patient.).” (L95-106)
Ø The paper discusses the generation of medical content by AI. How do the authors address the ethical considerations and potential misinformation risks associated with AI-generated medical content?
About ethical consideration:
To date, no AI tool is able to guarantee the confidentiality of the data as described below:
“4.4.3. Data privacy
Patient-related data should follow strict data privacy guidelines to avoid leaks or subsequent use for unrelated matters.
4.4.4. Political environment and regulations
Politically, GAIs have accelerated the discussions about their regulation, most notably in the European Union (EU). In effect, as proposed in the EU AI Act, GAIs would have to comply with transparency requirements such as (i) disclosing that the content was generated by AI, (ii) designing the model to prevent it from generating illegal content and (iii) publishing summaries of copyrighted data used for training.(43)
To date, no AI tool is able to guarantee the confidentiality of the data fed into its system. Since AIs are formatted to enrich themselves (and thus memorize and restitute) with the content they are trained on, they could divulge it. For the time being, therefore, it is illegal in Switzerland and Europe to feed AIs with personal data. Users must therefore take care to anonymize data transmitted to the AI(44).
Globally, in November 2023, a first international agreement was signed (Bletchley declaration) on GAI safety. Organized in London, this summit will be held once a year, bringing together experts and politicians to discuss the risks and regulation of AI.” (L403-419)
About misinformation:
Rigorous fact-checking and validation mechanisms should be implemented as described below:
“In order to avoid several shortcomings and pitfalls that were encountered, it is recommended to use both ChatGPT and Bard (as well as other similar GAI chatbots) responsibly and with strict guidelines (e.g. verification of sources, critical analysis of answers, awareness of risks pertaining to data privacy, etc).” (L423-426)
Ø The paper mentions the use of a committee, including medical students and specialists, to evaluate the AI responses. How was the evaluation process structured, and what criteria were used to assess the quality of the AI responses?
All questions and answers were copied and pasted into tables (see Appendices 1, 2 and 3). Responses were compared and discussed by a committee including two patients, two medical students and two specialists in plastic and reconstructive surgery. A general score was attributed to every answer based on quality (0 = no answer, 1 = poor, 2 = acceptable, 3 = good, 4 = very good; a star * after the score indicates that wrong or invented elements were suggested in the answers). (L111-116)
Ø The paper notes the limitations of AI in performing literature reviews and scientific writing. How do the authors envision the role of AI in academic research, given these limitations?
We think that future versions of AI models will include reliable, transparent and accurate literature reviews. The role of AI in academic research, therefore, might evolve as these models improve in their ability to access and analyze external sources. For now, AI will always be used as a tool in combination with expert knowledge and will not replace humans in the foreseeable future.
Ø The study explores the use of AI for providing postoperative care advice. How does the advice generated by AI compare to standard postoperative care guidelines, and what is the potential for AI to supplement or enhance patient care?
Standard postoperative care guidelines may vary from medical center to medical center, country to country, etc. It is thus beyond the scope of this paper to review such guidelines. That being said, general elements can be noted: full transparency of information to patients, full disclosure of information to patients, assurance that patients have fully understood the information provided to them, etc.
AIs are capable of learning based on the content provided to them. This may apply to the post-operative care of a patient depending on the measures in force in a care center. In addition, this would allow the patient to have the same information several times without having to see their doctor again as described below:
“As the authors suggest, one solution would be to create a chatbot on an online portal, edited directly by doctors, that would answer the questions of patients. Gupta et al. also posit that GAIs could be used to support the postoperative period between specialist appointments(12).” (L374-377)
Ø The paper discusses the potential for AI to save time and money in medical contexts. Can the authors provide more details on the specific areas where AI can offer the most significant cost and time efficiencies?
Students can use the AI quickly and without having to travel for their learning. The AI can serve as a relay between medical and nursing consultations to relieve the healthcare team and reassure the patient on subjects already discussed in consultation. For specialists, as discussed, AI can save time in administrative tasks.

Reviewer 2 Report
Comments and Suggestions for Authors
Artificial intelligence is undoubtedly the hottest research spot now. It has aroused widespread concern and discussion both among professionals and ordinary people. The purpose of this study is to compare and discuss the applications and limitations of ChatGPT and Bard in plastic surgery. The manuscript is an interesting study and provides some new ideas to the readers. It's a well-written manuscript. So, acceptance should be recommended for this manuscript.
Author Response
Dear Reviewer, thank you very much for your positive feedback regarding this study and we are glad it could provide new ideas to readers.
Reviewer 3 Report
Comments and Suggestions for Authors
The authors performed an audit of the answers given by two AI platform to certain plastic surgery related questions. The study aims are vague with aims to analyse risk -benefits of these platforms in plastic surgery. But it is not clear how are these aims achieved with the results they presented. The conclusion also did not specifically address this. All these needs to be clarified. The detailed comments are listed below:
1. Objective study not clear. Stated at the end of introduction to d identify opportunities while providing a risk-benefit analysis which is probably an aim. But this is not clear enough. I wold suggest the authors state clearly what are the aim of this study.
2. The analysis of risks / benefits were not clearly presented in the results. What are the analysed risk?
3. The issue of unclear objective becomes obvious at the end as there are no clear conclusion, as there are no clear objective to answer.
4. Table 1 to 3 has no description on it. Also, must explained the marks (already explained in text but must add here too) together with the descriptions
Author Response
Dear Reviewer, thank you for your meticulous proof-reading of the manuscript. Please find our answers to your questions below.
- Objectives of the study are twofold:
- Assess the capabilities of two AI tools that have gathered widespread attention and interest around the world in record time.
- Assess the relevance of these AIs to complete common tasks of three groups of persons in the medical field (students, specialists, patients) as few evaluations had been done to date. (L-80-84)
We there clarified the aims of the study and hope this will be satisfactory.
- The major assessed risk is the relevance of the answers provided by AI tools. In effect, with their rapid evolution and adoption, users might be tempted to use these AI tools without critically judging the relevance of answers that are provided to them. Here, by proving their capabilities, we highlight the limitations of two generative AIs while shining light on the opportunities they can provide when being prompted with specific tasks.
- In light of the perspectives outlined above (see Question 1), we believe that such studies help in assessing whether these AI tools should be used by medical students, plastic surgeons or patients in plastic surgery.
- Done.

Reviewer 4 Report
Comments and Suggestions for Authors
Carefully organized study and well prepared manuscript. It is ready for publication.
Author Response
Dear Reviewer, thank you very much for your positive feedback and favorable recommendation to publish this study.
Round 2
Reviewer 3 Report
Comments and Suggestions for Authors
Thank you for the revision based on the previous comments. However I would argue that the revision did not resolve the raised issues:
1. It is stated blatantly that this study will "provide risk-benefit analysis" of the use of these AI. But this is not the case of the conduct of the study and how the results were presented. Probably the authors should consider deleting such statement totally if no such analysis is presented.
2. The main objective should be one. The authors revised the objectives to two-fold which makes is more confusing. I would also suggest the authors consider revising the main objective to be more specific to the conduct of the study.
Author Response
Dear Reviewer,
Thank you for your second round of review and please find answers to your comments below. The changes are reflected in light blue in the PDF attached.
1. To avoid any misleading claims, we have simplified the below sentence to reflect the analysis which has been performed.
"Here, we report on the use of ChatGPT and Bard by these three groups of users and identify opportunities while providing an analysis for specific use cases." (L78-80)
2. In the previous round of review, we have tried to explicitly detail the steps leading to the main objective (hence twofold). We understand that this might be confusing, as you have pointed out, and have thus simplified the overall objective to have only one, as outlined in the below sentence:
"In this context, the objective is to assess the relevance of two Ais (ChatGPT and Bard) to complete common tasks of three groups of persons in the medical field outlined above as few comparative evaluations have been performed to date." (L80-84)
We thank you for your comments and are confident that these amendments will be bring more clarity to the manuscript and to our study in general.

Round 3
Reviewer 3 Report
Comments and Suggestions for Authors
thank you for the opportunity to review the manuscript. the authors have addressed all of the previous comments adequately. there are no additional comments from this reviewer.
Author Response
Thank you very much